# Synbiotics Alleviate Hepatic Damage, Intestinal Injury and Muscular Beclin-1 Elevation in Rats after Chronic Ethanol Administration

**DOI:** 10.3390/ijms222212547

**Published:** 2021-11-21

**Authors:** Yi-Hsiu Chen, Wan-Chun Chiu, Qian Xiao, Ya-Ling Chen, Hitoshi Shirakawa, Suh-Ching Yang

**Affiliations:** 1School of Nutrition and Health Sciences, Taipei Medical University, Taipei 11031, Taiwan; ma07108007@tmu.edu.tw (Y.-H.C.); wanchun@tmu.edu.tw (W.-C.C.); ma07106012@tmu.edu.tw (Q.X.); ylchen01@tmu.edu.tw (Y.-L.C.); 2Research Center of Geriatric Nutrition, College of Nutrition, Taipei Medical University, Taipei 11031, Taiwan; 3Laboratory of Nutrition, Graduate School of Agricultural Science, Tohoku University, Sendai 980-8857, Japan; shirakah@m.tohoku.ac.jp; 4Nutrition Research Center, Taipei Medical University Hospital, Taipei 11031, Taiwan

**Keywords:** alcoholic liver disease, Inulin-containing synbiotics, muscular protein metabolism

## Abstract

The purpose of this study was to investigate the beneficial effects of synbiotics on liver damage, intestinal health, and muscle loss, and their relevance in rats with chronic ethanol feeding. Thirty Wistar rats fed with a control liquid diet were divided into control and synbiotics groups, which were respectively provided with water or synbiotics solution (1.5 g/kg body weight/day) for 2 weeks. From the 3rd to 8th week, the control group was divided into a C group (control liquid diet + water) and an E group (ethanol liquid diet + water). The synbiotics group was separated in to three groups, SC, ASE, and PSE. The SC group was given a control liquid diet with synbiotics solution; the ASE group was given ethanol liquid diet with synbiotics solution, and the PSE group was given ethanol liquid diet and water. As the results, the E group exhibited liver damage, including increased AST and ALT activities, hepatic fatty changes, and higher CYP2E1 expression. Intestinal mRNA expressions of occludin and claudin-1 were significantly decreased and the plasma endotoxin level was significantly higher in the E group. In muscles, beclin-1 was significantly increased in the E group. Compared to the E group, the PSE and ASE groups had lower plasma ALT activities, hepatic fatty changes, and CYP2E1 expression. The PSE and ASE groups had significantly higher intestinal occludin and claudin-1 mRNA expressions and lower muscular beclin-1 expression when compared to the E group. In conclusion, synbiotics supplementation might reduce protein expression of muscle protein degradation biomarkers such as beclin-1 in rats with chronic ethanol feeding, which is speculated to be linked to the improvement of intestinal tight junction and the reduction of liver damage.

## 1. Introduction

The World Health Organization (WHO) reported that 3 million deaths each year were associated with the harmful use of alcohol [1]. Around 40~60% of patients with alcoholic liver disease (ALD) suffer from severe sarcopenia [2,3]. Possible mechanisms can be explained by the primary hit theory (gut-muscle axis) and the secondary hit theory (liver-muscle axis) [4]. For the primary hit, chronic alcohol intake induces dysfunction of the gut barrier and dysbiosis of the gut microbiota which might lead to endotoxemia which might induce muscle loss because of increasing protein degradation in [4]. Gut barrier dysfunction is linked to the damage of tight junction proteins, such as occludin, claudins, and zonula occludens (ZOs) [5]. It was found that messenger (m)RNA expressions of tight junction proteins were decreased after alcohol treatment both in animal and cell culture studies [6,7]. Moreover, the gut microbiota composition was turned into a composition of pathogenic bacteria accompanied by a decrease in bacterial diversity in rats chronically fed with alcohol [8,9,10]. On the other hand, based on the viewpoint of the gut-liver axis, ethanol-induced endotoxemia also impaired the metabolic ability of the liver, including ammonia detoxification [11]. It was demonstrated that hyperammonemia could be one of several potential reasons for abnormal muscle protein metabolism in ALD patients [12].

Skeletal muscle mass and protein content are maintained by the balance of protein synthesis and degradation [4]. For protein synthesis, the binding of insulin-like growth factor 1 (IGF-1) to its receptor activates phosphatidylinositol-3- kinase (PI3K), which is responsible for the phosphorylation of protein kinase B (Akt). Then, Akt activates the mammalian/mechanistic target of rapamycin (mTOR) and the downstream proteins such as ribosomal protein S6 kinase (S6K)1 (also known as p70S6 kinase, p70S6k) which promotes protein synthesis in muscle [13]. On the other hand, myostatin is expressed and secreted predominantly by skeletal muscle and functions as a negative regulator of muscle growth, because it activates the ubiquitin-proteasome pathway (UPP) and autophagy system which are involved in protein degradation in muscle [14]. Myostatin activates mothers against decapentaplegic homolog 2/3 (Smad2/3) and binds to Forkhead box protein O (FOXO), then regulates the expression of muscle atrophy F-box (atrogin-1) and muscle RING finger-1 (MuRF1), which contribute to protein degradation [14]. It was also reported that myostatin overexpression increased the expression of components of the autophagy pathway expression and turnover, including LC3-II and beclin-1, as well as autophagosome formation [15]. A previous study also reported that chronic alcohol consumption could lead to a decrease of phosphorylation of mTOR which initiated the autophagy system [16].

Synbiotics are combinations of probiotics and prebiotics which enhance the intestinal health of hosts [17]. It is known that probiotics have the effects of inhibiting the growth of harmful bacteria in the intestines, regulating the immune system and lipid metabolism, improving food digestion and absorption, and being an antioxidant [18]. Based on animal studies, it was found that synbiotic or probiotic supplementation improved liver function and maintained the structural integrity of the gut barrier in rodents with alcohol-induced liver injury [17,19]. According to our previous studies, it was found that supplementation with synbiotics might increase beneficial bacteria, restore the microbiota in the intestines, and reduce endotoxin transfer through gut tight junctions to ameliorate alcohol-induced liver injury [19]. However, the relationship between intestinal health and muscle loss has not been discussed in rats with chronic alcohol consumption. Therefore, the purpose of this study was to investigate the beneficial effects of synbiotics supplementation on the liver damage, intestinal health (including permeability and microbiota composition) and muscle loss in rats subjected to chronic ethanol feeding. Moreover, the relevance between intestinal health and muscle loss was also discussed when rats with ethanol-induced liver injuries were supplemented with synbiotics.

## 2. Results

In this study, thirty male Wistar rats were divided into five groups: C, E, SC, PSE, and ASE groups. The C group was fed the control liquid diet with distilled water for 8 weeks. The E group was fed the control liquid diet with distilled water for the first 2 weeks and then an ethanol liquid diet with distilled water for the next 6 weeks. The SC group was fed the control liquid diet with synbiotics solution (1.5 g/kg BW/day) for 8 weeks. The PSE group was fed a control liquid diet with synbiotics solution for the first 2 weeks and then an ethanol liquid diet with distilled water for the next 6 weeks. The ASE group was fed a control liquid diet with synbiotics solution for the first 2 weeks, and then an ethanol liquid diet with synbiotics solution for the next 6 weeks.

### 2.1. Food Intake and Food Efficiency

As shown in Table 1, no difference of food intake was found among the groups. Moreover, the ethanol intake showed no difference among the ethanol-fed groups (E, PSE, and ASE groups). The synbiotics intake did not differ when compared among the synbiotics-fed groups (SC, PSE and ASE groups). Compared to the C group, the E group showed significantly lower food efficiency. However, there was no difference in the food efficiency among the E, PSE, and ASE groups (Table 1).

### 2.2. Final BWs and Relative Liver Weights

Compared to the C group, the E group had a significantly lower final BW (Table 2). There were no differences in BWs among the E, PSE, and ASE groups. The E group had significantly higher liver weights and relative liver weights compared to the C group. The PSE and ASE groups had lower relative liver weights compared to the E group.

### 2.3. Liver Damage

#### 2.3.1. Plasma AST and ALT Activities, Ammonia Level, and Hepatic Histopathology Scores

As shown in Table 3, the E group had significantly higher ALT and AST activities and ammonia levels compared to the C group. The synbiotics-supplemented groups, such as the PSE and ASE groups, had significantly decreased ALT activity and ammonia levels, but there were no differences in AST activity compared to the E group (Table 3). The liver histological results were shown in Figure 1. Fatty changes, necrosis, and inflammation were observed in the E group, and the scores of necrosis and inflammation were significantly higher compared to the C group. Compared to the E group, the ASE group showed a significantly lower score of fatty changes, but only showed a decreasing trend of inflammation and necrosis (Figure 1).

#### 2.3.2. Hepatic Cytokines

The hepatic cytokine levels were shown in Table 4. There was no difference in each cytokine between the C and E groups. However, all cytokines was significantly decreased in the PSE group, but only IL-1β level was significantly reduced in the ASE group when compared to the E group (Table 4).

#### 2.3.3. Oxidative Stress

The TBARS levels in the plasma and liver were shown in Table 5. There was no difference between the E and C groups in plasma TBARS level. However, the E group had a significantly higher hepatic TBARS level compared to the C group. The hepatic TBARS levels of PSE and ASE groups were significantly reduced than that of E group (Table 5).

The protein expression of CYP2E1 was shown in Figure 2. Compared to the C group, the E group had significantly higher CYP2E1 protein expression in the liver. In contrast, the PSE and ASE groups showed there was significantly lower protein expression of CYP2E1, when compared to the E group (Figure 2).

### 2.4. Intestinal Damage

#### 2.4.1. Serum Endotoxin Level

As shown in Figure 3A, compared to the C group, the plasma endotoxin level was significantly elevated in the E group. The decreasing trend of plasma endotoxin level was observed in the PSE group, but a significantly lower endotoxin level was found in the ASE group, when compared to the E group (Figure 3A).

#### 2.4.2. Intestinal Tight Junction Protein mRNA Expressions

The tight junction mRNA expressions were shown in Figure 3B. In this study, mRNA expressions of claudin-1 and occludin were significantly lower in the E group compared to the C group. Compared to the E group, the PSE and ASE groups had significantly higher occludin and claudin-1 mRNA expressions (Figure 3B).

#### 2.4.3. Fecal Microbiotic Composition

##### Firmicutes (F) to Bacteroidetes (B) Ratio

The result of F/B ratio was shown in Figure 4. A lower F/B ratio was found in the E group, while it was not significant. The F/B ratio of the PSE and ASE groups was similar to that of the C group (Figure 4).

##### The α-Diversity and β-Diversity Indices

The result of α-diversity is shown in Figure 5A,B. The Chao-1 and ACE indices often refer to the richness of species, while the Shannon and Simpson indices indicate differences in species. There were no changes in the fecal microbiotic richness and differences between the C and E groups (Figure 5A). There were no differences in the Shannon and Simpson indices among all groups (Figure 5B). Figure 5C showed the result of β-diversity which is represented with the principal coordinate analysis (PCoA). According to the PCoA, the C, E, and SC groups showed different microbiotic distributions (Figure 5C).

##### Linear Discriminant Analysis of the Effect Size (LEfSe)

Comparing the specific bacterial taxonomical abundance differences between the C and E groups, pathogenic bacteria such as the Peptococcaceae (family), Bacillaceae (family), Acinetobacterlwoffii (genus), and Providencia (genus) were the dominant bacteria in the E group (Figure 6). Figure 7 showed the comparison of specific bacterial taxonomical abundances among the SC, PSE and ASE groups. The dominant bacteria of the SC group were Actinobactria (phylum), Actinobactria (class), Bifidobacteriaies (order), Bifidobacteriaceae (family), Actinomycetales (order), Bifidobacterium bifidum (genus), and Streptococcus (genus). The dominant bacteria of the ASE group contained Bifidobateriates (order), Bifidobacteriaceae (family), Streptococcus (genus), and Bifidobacterium longum (genus). Moreover, Acinetobacterlwoffii (genus) was found in the PSE group as the dominant bacteria (Figure 7). No specific dominant bacteria were found among the E, PSE, and ASE groups.

### 2.5. Muscle Loss

#### 2.5.1. Grip Strength and Muscle Histopathology

As shown in Table 6 and Figure 8, there were no significant differences of grip strength and CSA among all groups (Table 6, Figure 8).

#### 2.5.2. Muscle Protein Synthesis and Degradation

There were no differences among all groups in the protein synthesis markers such as mTOR and S6K protein expressions (Figure 9). As the protein degradation, there were no differences in the protein expressions of myostatin, MuRF-1, and LC3BII/I protein expressions among all groups (Figure 9). However, beclin-1 as the autophagy marker was significantly higher in the E group compared to the C group and was significantly decreased in the PSE and ASE groups compared to the E group.

### 2.6. Amino Acid Composition

#### 2.6.1. Amino Acid Levels in Plasma

Compared to the C group, the E group had higher branched-chain amino acid (BCAA) levels and significantly higher phenylalanine, aromatic amino acid (AAA), methionine, histidine, alanine, and proline levels in plasma. Furthermore, the E group had a significantly lower arginine level compared to the C group. The PSE and ASE groups had significantly lower phenylalanine, AAA, methionine, histidine, alanine, and glutamine levels compared to the E group (Appendix A).

#### 2.6.2. Amino Acid LEVELS in Liver

Appendix A showed the amino acid levels in liver. The E group had a significantly lower glutamic acid level and a significantly higher proline level in liver when compared to the C group. There were no differences in hepatic amino acid levels among the E, PSE, and ASE groups.

#### 2.6.3. Amino Acid Levels in Muscles

As shown in Appendix A, compared to the C group, the SC group had significantly higher phenylalanine, AAA, methionine, lysine, and glutamic acid levels in muscles. However, there were no differences in all amino acid levels in muscles among the C, E, PSE, and ASE groups.

## 3. Discussion

### 3.1. Food Intake, Alcohol Intake, and Synbiotics Intake

There were no differences in food intake levels and total calories among all groups (Table 1). The alcohol intake in the alcohol-feeding group was 3.8~4.1 g/day, which was equal to 97 g/day in a 60 kg human. It has been reported that drinking over 60 g of ethanol/day could be defined as heavy alcohol consumption in humans [20].

### 3.2. BWs and Relative Liver Weights

Ethanol-fed rats had lower BWs and higher relative liver weights in this study (Table 2) which was similar with our previous studies [19,21]. Previous studies reported that chronic alcohol consumption affected nutrient absorption through damage to the intestines [22]. Moreover, alcohol only provides calories but not nutrients, and this might also have led to a decreased BW. Chronic alcohol consumption might cause an increase in the relative liver weight due to abnormal lipid and protein metabolism [23]. However, synbiotics supplementation could decrease the relative liver weight in rats chronically fed with alcohol in this study (Table 2, E vs. PSE and ASE groups).

### 3.3. Liver Damage

In the present study, it was observed that plasma AST and ALT activities, ammonia levels, and hepatic fatty changes had increased in the ethanol-fed groups (Table 3 and Figure 1, C vs. E groups), which indicated that alcohol-induced liver damage was successfully established in this study [21,24]. Moreover, protein expressions of CYP2E1 and TBARS level in the liver were also significantly increased in the E group (Table 5 and Figure 2, C vs. E groups). When alcohol is metabolized by CYP2E1, a large amount of reactive oxygen species (ROS) is generated, such as superoxide anions (O_2_^−^) and hydroxyl radicals (∙OH) [25]. Excessive ROS can induce lipid peroxidation, protein damage, DNA impairment, and cell damage [26]. Therefore, it could be considered that oxidative stress was also induced in rats after chronic alcohol feeding in this study. However, there was only a higher trend of hepatic cytokines in the E group in the present study (Table 4) which might due to the shorter feeding period compared to our previous studies [19,21].

When supplemented with synbiotics, liver damage were inhibited including lower AST and one might speculate that synbiotics supplementation could inhibit ethanol-induced liver damage including lower plasma ALT activity, ammonia level, hepatic fatty changes, and CYP2E1 protein expression (Table 3, Figure 1 and Figure 2). In the ethanol-fed animal models, it was found that supplementation with probiotics or prebiotics could decrease AST and ALT activities and suppress plasma ammonia levels [27,28]. The possible reason might be linked with the intestinal protection of probiotics or prebiotics, thus the related factors of intestinal health was also discussed in this study as below.

### 3.4. Intestinal Damage

The present study showed that rats with ethanol feeding had the lower mRNA expression of occludin and claudin-1, and had higher serum endotoxin level (Figure 3, C vs. E groups). Epithelial tight junctions confer a barrier function in the intestinal mucosa by forming a diffusion barrier to toxins, allergens, and pathogens from the gut lumen to tissues and the systemic circulation [29,30]. Previous studies indicated that mRNA expressions of occludin and claudin-1 were decreased after alcohol treatment [6,7]. Chronic alcohol consumption also caused intestinal inflammation due to ethanol and acetaldehyde, changed the microbiotic composition, and disrupted the intestine immune system [31]. Thus, it was indicated that chronic alcohol feeding could impair the intestinal tight junction proteins which allowed endotoxin to enter the blood stream. This study also observed that the synbiotics treatment could elevated the mRNA expressions of occludin and claudin-1, and reduced the plasma endotoxin level in rats chronically fed with alcohol (Figure 3, E vs. PSE and ASE groups). It has been found that probiotics treatment could improve the intestinal permeability and maintain intestinal tight junctions, and thereby ameliorating intestinal inflammation [6,27,32]. Therefore, the protective effect of synbiotics on the tight junction structure in the ileum was also defined in rats fed with ethanol-containing liquid diet in this study.

In the present study, there was no differences in the F/B ratio among all groups, only the SC group had the highest F/B ratio, which might to be linked to synbiotics supplementation (Figure 4). In previous studies, there was no concordance of the F/B ratio in alcohol-fed animals [33,34,35] because of the different administration route, feeding duration or various animal species etc.

On the other hand, the present study showed that the E group had more pathogenic bacteria in the fecal microbiota, such as the Peptococcaceae (family), Bacillaceae (family), Acinetobacterlwoffii (genus), and Providencia (genus) (Figure 6). Ponziani et al. reported that the bacteria in the Clostridia class, such as the Clostridiaceae, Ruminococcaceae, Veillonellaceae, Lachnospiraceae, Peptostreptococcaceae, and Peptococcaceae families, were the most abundant microorganisms in cirrhosis patients [36]. Moreover, Peptococcaceae family was considered as the endotoxin-producing bacteria [36]. Przewłócka et al. reported that losses of muscle mass and strength were related to a decrease in Bifidobacterium and increase in the Enterobacteriaceae and endotoxin-producing bacteria [37]. Bacillus, a genus in the Bacillaceae family, was reported to be a potential pathogen which caused food-borne diseases and infections in immunocompromised individuals [38]. Shah et al. also reported that the Providencia infection was often related to diarrhea due to invasion of the intestinal mucosa [39]. In addition, Acinetobacterlwoffii can cause nosocomial infections such as pneumonia, meningitis, urinary tract infections, wound infections, and gastroenteritis [40]. According to previous studies, the present study indicated that rats under chronic ethanol ingestion showed lower levels of anti-inflammatory bacteria and the gut microbiota composition which easily produce endotoxin. On the other hand, dominant bacteria in the SC and ASE groups contained supplemented probiotics, including Actinobactria (phylum), Actinobacrtria (class), Bifidobacteriaies (order), Bifidobacteriaceae (family), Actinomycetales (order), Bifidobacteriumbifidum (genus), and Streptococcus (genus). Moreover, Acinetobacterlwoffii (genus) was the dominant bacteria in the PSE group (Figure 7). Based on these results, it was found that the gut microbiota composition of ASE group was similar with that of the SC group. To summarize the above results, synbiotics supplementation might maintain intestinal health through preventing tight junction damage and maintaining a healthier microbiota composition in rats chronically fed with alcohol.

### 3.5. Muscle Protein Metabolism and Liver-Gut-Muscle Axis

The present study found that only muscular beclin-1 protein expression was significantly increased in the E group (Figure 9). Nearly 40~60% of cirrhosis patients experience muscle loss, particularly patients with alcoholic cholestatic cirrhosis whose muscle wasting is the most severe [41]. Animal studies also indicated that chronic alcohol consumption reduced muscle protein synthesis through decreasing the protein expression of mTOR and downstream factors, such as S6K and 4EBP1 [42]. In chronic alcohol feeding animals, higher proteins expressions of degradation and autophagy were observed including Murf-1, antrogin-1, LC3B-II, p62, and beclin-1 [4,43]. In addition, it was found that ethanol-treated C2C12 myotubes showed increased protein expression of LC3B-II [44]. Therefore, the present results were similar with previous studies in the reduced protein expression of beclin-1 because of ethanol intake. On the other hand, high plasma endotoxin and ammonia levels inhibited the phosphorylation of Akt and mTOR and increased the protein expressions of myostatin and MuRF-1 which might be the reason for the elevation of muscle protein degradation [37,45,46]. In this study, rats fed with ethanol indeed showed the higher plasma endotoxin level (intestine damage) and ammonia level (liver damage), which might be connected with the intestine and liver injuries because of chronic ethanol consumption. However, the other biomarkers of muscle loss, such as muscle CSA and grip strength, did not show a difference among groups in the present study. Extending the experimental period is necessary in the further study.

In this study, muscular beclin-1 protein expression was significantly decreased in the synbiotics-supplemented groups (Figure 9, E vs. PSE and ASE groups). This result represented that synbiotics supplementation might have a potential effect of maintaining the muscle mass which also connected with the lower plasma endotoxin and ammonia levels in rats fed with ethanol and synbiotics. Other relevant parameters in muscle tissue should be evaluated in future study, such as measure proteasome chymotrypsin-like activity (proteasome-ubiquitin protein degradation) and the levels of lysosome-associated membrane glycoprotein-1 (LAMP-1, autophagic flux).

### 3.6. Amino Acid Composition

The plasma amino acid composition is affected by several factors, including chronic alcohol consumption, dietary protein deficiencies, and the severity of liver disease [47]. It has been reported that liver cirrhosis might cause abnormal BCAA concentrations in plasma [46,48] which led to muscle protein degradation [46]. In contrast, Shaw et al. reported that chronic alcohol consumption in baboons produced higher plasma BCAA levels released from muscle due to the impaired metabolism of BCAAs in muscles [47]. In the present study, no change of plasma BCAA levels was found in the E group (Appendix A). The possible reason might be related to the lower severity of liver disease because the hepatic fibrosis were not observed in rats fed with alcohol in this study (data not shown).

AAA levels, such as phenylalanine and tyrosine, were elevated in cirrhosis patients, possibly due to the muscle catabolism. Moreover, AAAs and methionine are primarily catabolized in the liver, and impairment of liver function could lead to an increase of AAAs and methionine levels in plasma [47,48,49]. Phenylalanine and tyrosine could be indicators of proteolysis, while these two amino acids cannot be reutilized for protein synthesis [50]. The present study indicated that plasma AAAs and methionine levels were higher in the E group which was connected with the impaired liver function and muscle proteolysis in rats under chronic ethanol feeding (Appendix A). Moreover, PSE and ASE groups had significantly lower plasma methionine levels compared to the E group (Appendix A). This result possibly indicated that synbiotics supplementation normalized the metabolism of AAAs and methionine due to ameliorate ethanol-induced liver injury.

Proline and hydroxyproline are associated with collagen synthesis [51]. The reaction of proline synthesis from ornithine and glutamic acid is reversible through the intermediate mitochondrial metabolite [51]. Mata et al. reported that alcoholic liver cirrhosis and hepatitis patients had higher serum proline and hydroxyproline levels [52]. In the present study, higher proline levels in the plasma and liver were also found; however, the glutamic acid level was lower in the liver in the E group (Appendix A). These results indicated that chronic alcohol feeding disrupted the metabolism of proline and glutamic acid, which might diminish collagen synthesis.

### 3.7. The Study Limitation

This study still has several limitations. The measurement of the endotoxin/TLR4 signaling pathway in the muscle tissue is necessary in order to confirm the anti-inflammation ability by synbiotics supplementation in rats fed with ethanol, because the plasma endotoxin level was decreased in synbiotics supplementation groups. Moreover, this study did not clearly reveal a difference in the protective effects between pre-administration (PSE group) and administration from start to the end (ASE group). In the future studies, it is important to discuss the therapeutic effects of synbiotics for rats that already display ethanol-induced damage. Additionally, the different dosage of synbiotcs supplementation should also be discussed in a further study.

## 4. Materials and Methods

### 4.1. Animals

Thirty 8-week-old male Wistar rats (BioLasco Taiwan, Ilan, Taiwan) were used in this study. Rats were caged individually at 23 ± 2 °C and 55% ± 10% relative humidity with a 12 h dark/light cycle. Animal experiments were approved by the Institutional Animal Care and Use Committee of Taipei Medical University (LAC-2017-0384).

### 4.2. Study Protocol

The Lieber and DeCarli ethanol liquid diet contains 36% of calories from 99% ethanol, and in order to match the calorie level, maltodextrin was used to replace ethanol for the control liquid diet. The inulin-containing synbiotic powder (FloraGuard^®^, Viva Life Science, Costa Mesa, CA, USA) (1.5 g/kg body weight (BW)/day) was dissolved in 10 mL distilled water and given by a small drinking tube every day. The components of synbiotic powder are shown in Table 7.

After 1 week of adaptation, rats were divided into a control group (*n* = 12) and a synbiotics group (*n* = 18). These two groups were given the control liquid diet and provided with distilled water (control group) or a synbiotics solution (synbiotics group, 1.5 g/kg BW/day) for 2 weeks. In the 3rd week of the experiment, the control group was divided into two groups (*n* = 6 in each group), one was still given the control liquid diet (C group) and the other one was given an ethanol liquid diet (E group). The synbiotics group was divided into three groups (*n* = 6 for each group). One was given the control diet with the synbiotics solution (SC group), another group was given the ethanol liquid diet with the synbiotics solution (ASE group), and the final group was given the ethanol liquid diet with distilled water (PSE group). During the experimental period, all groups were isoenergetic pair-fed based on the E group. Diet and water consumption levels were recorded every day, and the BW was recorded every week. Rats were anesthetized and sacrificed at the end of 8th week. Blood samples were collected via the ventral aorta into heparin-containing tubes (Becton Dickinson, Franklin Lakes, NJ, USA) and centrifuged at 1200× *g* for 15 min at 4 °C, and then plasma was collected and stored at −80 °C until analysis. Liver, muscle, and intestinal tissues were collected and stored at −80 °C for further analysis.

### 4.3. Measurements and Analytical Procedures

#### 4.3.1. Indicators of Liver Injury

##### Liver Function Index and Ammonia Level

Plasma biochemical parameters including aspartate aminotransferase (AST) and alanine aminotransferase (ALT) activities were analyzed with the ADVIA Chemistry XPT system (Siemens Healthcare Diagnostics, Eschborn, Germany). The plasma ammonia level was analyzed with the Dimension EXL 200 Integrated Chemistry System (Siemens Healthcare Diagnostics, Eschborn, Germany).

##### Histological Examinations

Liver tissues were fixed in a 10% formaldehyde solution and embedded in paraffin. Liver paraffin sections were cut and stained with hematoxylin and eosin (H&E) and trichrome stains. Experienced pathologists semi-quantitatively histologically evaluated liver specimens according to the degree of tissue damage, which was scored on a scale of 0 = absent, 1 = trace, 2 = mild, 3 = moderate, and 4 = severe [19].

##### Hepatic Cytokines

Liver tissues were cut into 0.5 g pieces and homogenized in 1.5 mL of iced homogenization buffer which contained 50 mM Tris (pH 7.2), 150 mM NaCl, 1% Triton-X, and 0.1% protease inhibitor (HYK0010, MedChemExpress, Monmouth Junction, NJ, USA), and then centrifuged at 3000× *g* and 4 °C for 15 min. Levels of tumor necrosis factor (TNF)-α, interleukin (IL)-1β, IL-6, and IL-10 were analyzed by commercial enzyme-linked immunosorbent assay (ELISA) kits from R&D Systems (Minneapolis, MN, USA). Supernatants of homogenized solutions were analyzed according to the assay kit instructions. A microplate reader (Molecular Devices, Sunnyvale, CA, USA) was used to read the optical density (OD) at 450 nm for all cytokines.

##### Plasma and Hepatic Lipid Peroxidation

One gram of liver tissue was homogenized with buffer containing 50 mM Tris (pH 7.2), 150 mM NaCl, 1% Triton-X, and 0.1% protease inhibitor (HYK0010, MedChemExpress, Monmouth Junction, NJ, USA), and then centrifuged at 3000× *g* and 4 °C for 15 min. Supernatants of the liver homogenate and plasma were used to analyze lipid peroxidation. Lipid peroxidation in plasma and liver homogenates was analyzed by thiobarbituric acid-reactive substance (TBARS) levels, using a TBARS kit from Cayman (TBARS 10009055 (TCA Method) Assay Kit, Cayman Chemical, Ann Arbor, MI, USA). The analytical procedure was according to the assay kit instructions.

##### Protein Expression of Hepatic Cytochrome P450 2E1 (CYP2E1)

CYP2E1 quantification in liver tissue was performed using a Western blotting method based on procedures described by Uesugi et al. and Chen et al. [53,54]. The antibodies were listed in Table 8.

#### 4.3.2. Assessment of Intestinal Damage

##### Serum Endotoxin Level

The endotoxin level was analyzed with a Pyrochrome Limulus Amebocyte Lysate Kit (Associates of Cape Cod, East Falmouth, MA, USA). Samples were diluted 4X before the experiment. A sample at 50 μL was mixed with the Pyrochrome assay buffer, and then incubated at 37 °C in the absence of light. The reaction was checked every 5 min, and the total reaction time was about 20~30 min. Acetic acid (50%) was added to stop the reaction. An ELISA reader (Multiskan RC, Lab Systems, Vantaa, Finland) was used to read the absorbance at OD 405 nm.

##### The mRNA Expressions of Intestinal Tight Junction

Total RNA of the ileum was extracted with TRI Reagent^®^ (Sigma-Aldrich, St. Louis, MO, USA), according to the instructions from the vendor. The concentration of RNA was calculated by the ratio of OD 260/280 and was read on a BioTek epoch reader with the Gen5TM Take3 Module (BioTek Instruments, Winooski, VT, USA). The concentration of total RNA was adjusted to 4000 ng/μL, and total RNA was reverse-transcribed with a RevertAid First Strand cDNA Synthesis kit (#K1621, ThermoFisher Scientific, Waltham, MA, USA). The concentration of complementary (c)DNA was adjusted to 50 ng/μL based on results from the BioTek epoch reader with the Gen5TM Take3 Module system. The resulting cDNA was amplified in a 96-well polymerase chain reaction (PCR) plate with SYBR Green/ROX qPCR Master Mix (2X) (ThermoFisher Scientific, Waltham, MA, USA) on a QuantStudio 1 Real-Time PCR System (ThermoFisher Scientific, Waltham, MA, USA). The primer sequences are listed in Table 9. Gene levels were normalized to β-actin, and all groups were compared to the C group by setting the C group to 1.

##### Fecal Microbiotic Composition

At the end of the 8th week, fresh feces was directly collected into sterilized 2 mL Eppendorf tubes and then stored in a −80 °C refrigerator until being analyzed. The fecal microbiotic distribution was analyzed with a 16s RNA Next Generation Sequencing (NGS) system, using the Greengenes database for classification. Agencourt AMPure XP Reagent beads (Beckman Coulter, Brea, CA, USA) were used to purify the amplified DNA. A quantitative (q)PCR (KAPA SYBR FAST qPCR Master Mix) was used to quantify each library in the Roche LightCycler 480 system (Roche, Basel, Switzerland). Then, each sample was diluted to 4 nM for the Illumina MiSeq NGS system (Illumina, Foster, CA, USA). The QIIME package tool was used for bioinformatics microbiome analyses, including operational taxonomic unit (OTU) assignment, phylogenetic reconstruction, diversity analyses, and visualization. OTUs were assigned for each sequence based on 97% similarity.

#### 4.3.3. Assessment of Muscle Loss

##### Grip Strength

The forelimb grip strength was evaluated with an animal forelimb grip strength measuring device (Model-RX-5, Aikoh Engineering, Nagoya, Japan) at the baseline and the 8th week.

##### Histological Examination

Right gastrocnemius muscle paraffin sections were cut and stained with H&E stain. The area of myofibers was used for counting the cross-sectional area (CSA). The CSA was counted and quantitated by Image-Pro Plus Software (Media Cybernetics, Rockville, MD, USA).

##### Protein Expressions of Protein Synthesis and Degradation

Right gastrocnemius muscle tissues were cut into 0.1 g pieces and homogenized in 0.4 mL RIPA buffer (50 mM Tris-HCl, 150 mM NaCl, 0.1% SDS, and 1% NP-40 at pH 7.5) containing 1% of a PI (HYK0010, MedChemExpress, Monmouth Junction, NJ, USA) and phosphatase inhibitor (PPI) (HYK0022, MedChemExpress, Monmouth Junction, NJ, USA). After an ice bath for 30 min, the homogenate was centrifuged at 10,000× *g* for 10 min at 4 °C. The target proteins were analyzed by western blotting described as the previous studies [53,54]. The antibodies were listed in Table 8.

#### 4.3.4. Amino Acid Composition of Plasma, Liver, and Muscle Tissues

The method of extracting amino acids from plasma, liver, and muscle tissues was modified from the protocol of Yuan et al. [55]. The amino acid analysis was performed by UPLC on an Acquity UPLC System coupled with the Xevo TQ MS system (Waters, Milford, MA, USA). For UPLC, a 1.7 mm (2.13100 mm) C18 column (Waters, Milford, MA, USA) was used. LC separation was carried out at 40 °C with a flow rate of 0.3 mL/min using the following gradient for the analysis: 0~0.5 min 1% B, 0.5~2.5 min from 1% B to 10% B, 2~3.5 min from 10% to 35% B, 3.5~6 min from 35% to 99% B, and 6~9 min 1% B [solvent system A: water/formic acid (100:0.1, vol/vol); B: acetonitrile/formic acid (100:0.1, vol/vol)]. Data were acquired by TargetLynx software (Waters, Milford, MA, USA).

### 4.4. Statistical Analysis

Data are presented as the mean ± standard deviation (SD) and analyzed by Statistical Analysis System (SAS, vers. 9.4, SAS Institute, Cary, NC, USA). Statistical differences were analyzed through a one-way analysis of variance (ANOVA) with Duncan’s post-hoc test for comparisons. Differences were considered statistically significant at *p* < 0.05.

## 5. Conclusions

The results of this study indicated that synbiotics supplementation decreased the muscular protein expression of beclin-1, which was a central autophagic regulator when rats were fed with ethanol-containing liquid diet. It was speculated that the reduction of muscular beclin-1 by synbiotics supplementation might be due to two major pathways (Figure 10). One is that synbiotics might be linked to a strengthening of intestinal tight junctions, thereby inhibiting endotoxin into the blood circulation. The other one is that not only the plasma ammonia level was reduced but also the AAAs metabolism was normalized due to the improvement of hepatic metabolic ability by synbiotics supplementation.

## Figures and Tables

**Figure 1 ijms-22-12547-f001:**
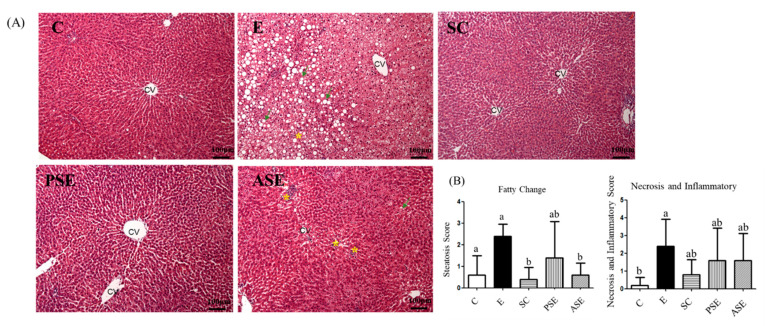
Effects of synbiotics on the histological examination of liver in rats with chronic ethanol feeding. (**A**) hematoxylin A and eosin (H&E) staining of liver pathological changes. (**B**) Quantification of fatty changes and inflammatory cells. Data are expressed as the mean ± standard deviation (*n* = 5). Different letters (a and b) indicate a significant difference at *p* < 0.05 by a one-way ANOVA with Duncan’s post-hoc test. Arrows indicate fatty droplets, while * indicates inflammatory cell in-filtration. CV, central vein.

**Figure 2 ijms-22-12547-f002:**
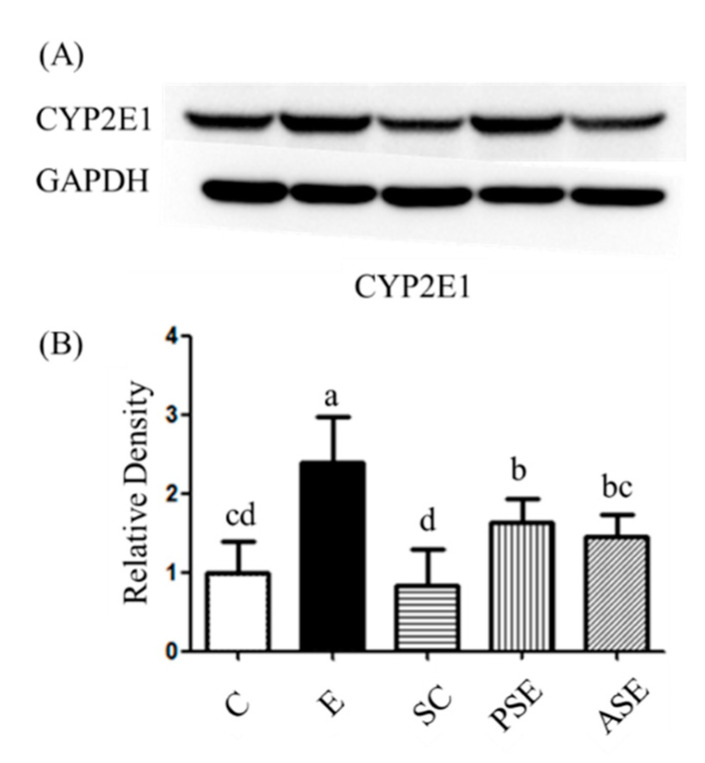
Effects of synbiotics on hepatic cytochrome P 450 2E1 (CYP2El) protein expression in rats with chronic ethanol feeding. (**A**) Western blot analysis of CYP2E1 protein expression. Glyceraldehyde 3-phosphate dehydrogenase (GAPDH) was used as internal control. (**B**) Quantitative analysis of protein levels and the ratio of each internal control was calculated by setting the value of the mean of the C group. Bars are the mean ± standard deviation (*n* = 6). Different letters indicate a significant difference at *p* < 0.05 by a one-way ANOVA with Duncan’s post-hoc test.

**Figure 3 ijms-22-12547-f003:**
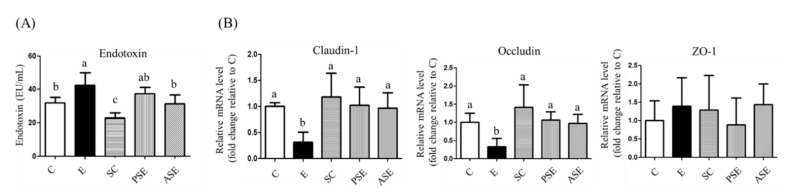
Effects of synbiotics on plasma endotoxin level and intestinal mRNA expressions of claudin-1, occludin, and zonula occludens (ZO)-1 in rats with chronic ethanol feeding. (**A**) Plasma endotoxin level. (**B**) Intestinal mRNA expressions of claudin-1, occludin, and zonula occludens (ZO)-1. β-Actin was used as an internal control. mRNA levels were calculated by the 2-ΔΔCT method, and the ratio of each internal control was calculated by setting the value of the mean of the C group to 1. Bars are the mean ± standard deviation (*n* = 6). Different letters indicate a significant difference at *p* < 0.05 by a one-way ANOVA with Duncan’s post-hoc test.

**Figure 4 ijms-22-12547-f004:**
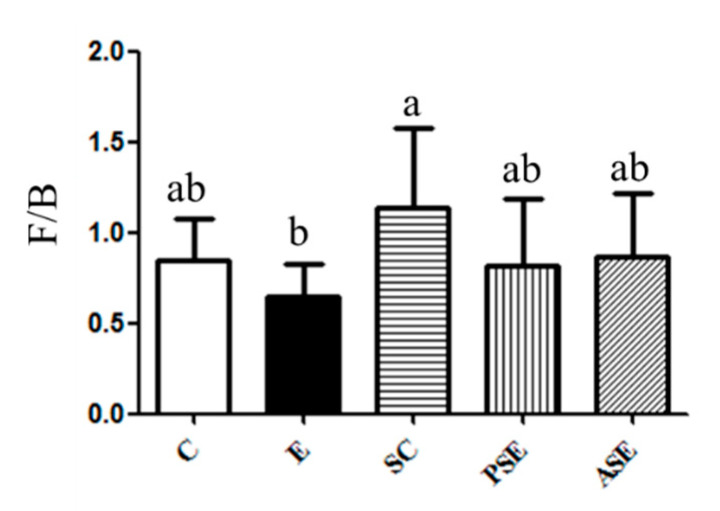
Effects of synbiotics on the Firmicutes-to-Bacteroidetes (F/B) ratio in rats with chronic ethanol feeding. Bars are the mean ± standard deviation (*n* = 6). Different letters indicate a significant difference at *p* < 0.05 by a one-way ANOVA with Duncan’s post-hoc test.

**Figure 5 ijms-22-12547-f005:**
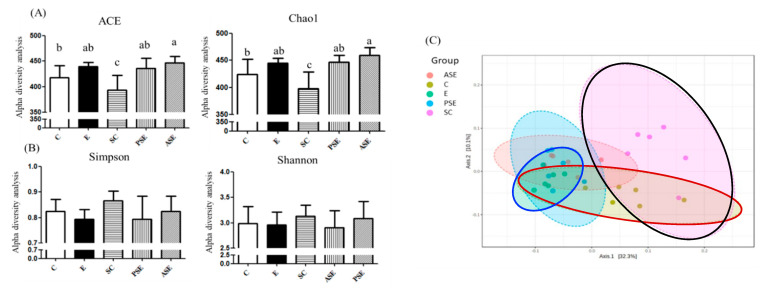
Effects of synbiotics on the α-diversity and principal coordinate analysis (PCoA) of the fecal microbiota in rats with chronic ethanol feeding. (**A**) Richness of the fecal microbiota. (**B**) Diversity of the fecal microbiota. (**C**) Principal coordinate analysis (PCoA) of the fecal microbiota. Bars are the mean ± standard deviation (*n* = 6). Different letters indicate a significant difference at *p* < 0.05 by a one-way ANOVA with Duncan’s post-hoc test. The areas inside the red, blue and black lines represent the microbiota distribution of the C, E and SC groups, respectively.

**Figure 6 ijms-22-12547-f006:**
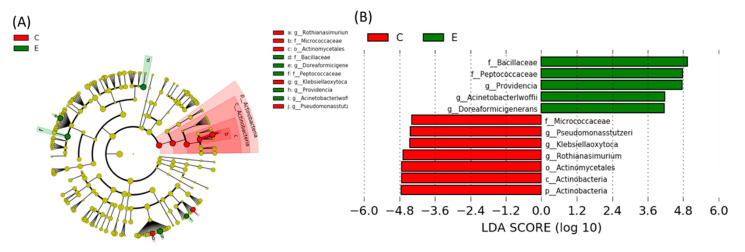
Effects of synbiotics on taxonomies of the fecal microbiotic compositions in rats with chronic ethanol feeding. (**A**) A linear discriminant analysis of the effect size (LEfSe) of the most significant abundance differences in the fecal microbiota between the C and E groups (*n* = 6). (**B**) Bacteria meeting the linear discriminant analysis (LDA) threshold (≥2.5) (*n* = 6).

**Figure 7 ijms-22-12547-f007:**
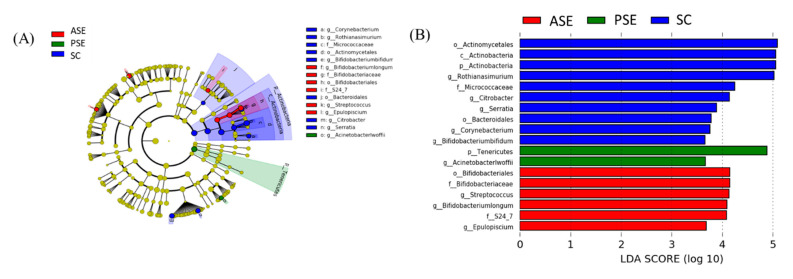
Effects of synbiotics on taxonomies of the fecal microbiotic compositions in rats with chronic ethanol feeding. (**A**) A linear discriminant analysis of the effect size (LEfSe) of the most significant abundance differences in the fecal microbiota among the SC, PSE, and ASE groups (*n* = 6). (**B**) Bacteria meeting the linear discriminant analysis (LDA) threshold (≥2.5) (*n* = 6).

**Figure 8 ijms-22-12547-f008:**
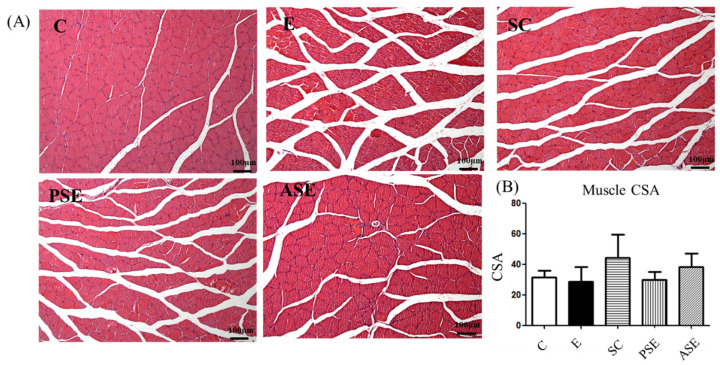
Effects of synbiotics on muscle cross-sectional area (CSA) in rats with chronic ethanol feeding. (**A**) H&E staining of pathological changes in the right gastrocnemius muscle. (**B**) Quantification of the area of myofibers, counted by the CSA. Bars are the mean ± standard deviation (*n* = 5). Different letters indicate a significant difference at *p* < 0.05 by a one-way ANOVA with Duncan’s post-hoc test.

**Figure 9 ijms-22-12547-f009:**
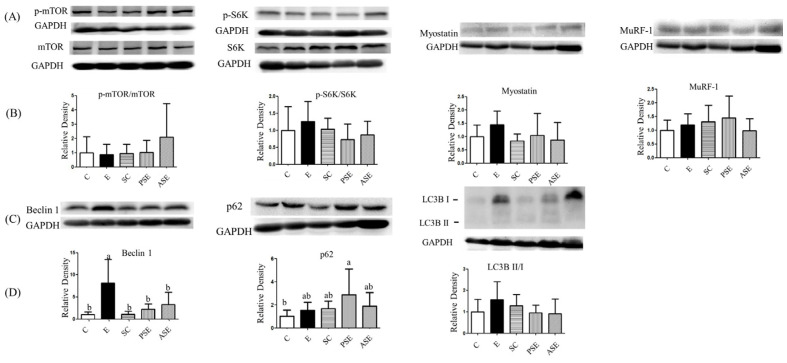
Effects of synbiotics on the related factors of muscle protein synthesis and degradation in rats with chronic ethanol feeding. (**A**) Western blot analysis of p-mTOR/mTOR, p-S6K/S6K, myostatin and MuRF-1 protein expressions. Glyceraldehyde 3-phosphate dehydrogenase (GAPDH) was used as an internal control. (**B**) Quantitative analysis of protein levels and the ratio of each to the internal control was calculated by setting the value of the mean of the C group to 1. (**C**) Western blot analysis of beclin-1, p62 and LC3BII/I protein expressions. Glyceraldehyde 3-phosphate dehydrogenase (GAPDH) was used as an internal control. (**D**) Quantitative analysis of protein levels and the ratio of each to the internal control was calculated by setting the value of the mean of the C group to 1. Bars are the mean ± standard deviation (*n* = 6). Different letters indicate a significant difference at p < 0.05 by a one-way ANOVA with Duncan’s post-hoc test.

**Figure 10 ijms-22-12547-f010:**
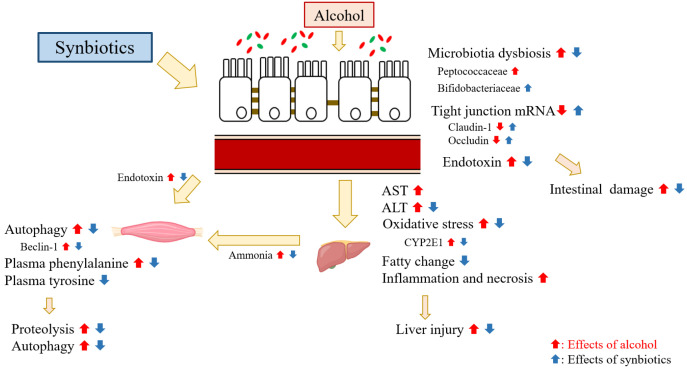
Effects of synbiotics supplementation on inhibition of muscle degradation in rats with chronic ethanol feeding. (1) In this study, it was indicated that synbiotics inhibited the protein degradation in muscle tissue when rats fed with an ethanol-containing diet. (2) The inhibition of muscular protein degradation by synbiotics supplementation may be due to two underlying pathways. First, when rats chronically ingested ethanol, synbiotics maintained the intestinal integrity and well-balanced microbiota composition, thereby reducing the lower plasma endotoxin level, which inhibited muscle damage. Secondly, synbiotics inhibited the elevation of plasma ammonia level and regulated the AAAs metabolism by ameliorating ethanol-induced liver injury based on the gut-liver axis, then reduced the protein degradation in muscle.

**Table 1 ijms-22-12547-t001:** Effects of synbiotics supplementation on the intake of food, ethanol and synbiotics in rats with chronic ethanol feeding ^1^.

Groups	Food Intake (g) ^2^			Ethanol Intake (g)			Synbiotics Intake (g/kg BW)			Food Efficiency ^3^ (%)			
C	81.3	±	0.8	-			-			4.3	±	0.2	^ab^
E	80.7	±	2.7	4.0	±	0.1	-			3.8	±	0.4	^c^
SC	81.2	±	0.8	-			1.4	±	0.1	4.5	±	0.3	^a^
PSE	78.6	±	1.9	3.9	±	0.1	1.4	±	0.7	4.0	±	0.4	^bc^
ASE	80.1	±	1.1	4.0	±	0.1	1.6	±	0.1	3.9	±	0.5	^bc^

^1^ Data are expressed as the mean ± standard deviation (*n* = 6). Different letters (^a^, ^b^, and ^c^) indicate a significant difference at *p* < 0.05 by a one-way ANOVA with Duncan’s post-hoc test. ^2^ 1 g = 1 kcal. ^3^ Food efficiency = (body weight gain/food intake) × 100.

**Table 2 ijms-22-12547-t002:** Effects of synbiotics supplementation on body weight (BW) and liver weight in rats with chronic ethanol feeding ^1^.

Groups	Final Body Weight (g)				Liver Weight (g)				Relative Liver Weight (%)			
C	418.2	±	14.1	^ab^	10.5	±	1.3	^b^	2.5	±	0.3	^cd^
E	389.8	±	21.4	^c^	12.9	±	1.5	^a^	3.3	±	0.3	^a^
SC	424.7	±	13.0	^a^	10.4	±	0.5	^b^	2.5	±	0.1	^d^
PSE	398.4	±	26.3	^bc^	10.8	±	0.9	^b^	2.7	±	0.1	^bc^
ASE	397.6	±	18.8	^bc^	10.9	±	0.4	^b^	2.7	±	0.1	^b^

^1^ Data are expressed as the mean ± standard deviation (*n* = 6). Different letters (^a^, ^b^, and ^c^) indicate a significant difference at *p* < 0.05 by a one-way ANOVA with Duncan’s post-hoc test.

**Table 3 ijms-22-12547-t003:** Effects of synbiotics supplementation on plasma aspartate aminotransferase (AST) and alanine aminotransferase (ALT) activities and ammonia (AMM) levels in rats with chronic ethanol feeding ^1^.

Groups	AST (U/L)				ALT (U/L)				AMM (μg/dL)			
C	70.0	±	8.6	^b^	45.0	±	8.5	^c^	63.4	±	12.5	^bc^
E	139.5	±	32.4	^a^	120.3	±	24.9	^a^	104.4	±	52.1	^a^
SC	82.0	±	8.9	^b^	46.7	±	4.3	^c^	37.0	±	11.0	^bc^
PSE	122.8	±	48.5	^a^	80.5	±	15.3	^b^	20.5	±	5.3	^c^
ASE	134.8	±	25.5	^a^	76.8	±	6.2	^b^	26.5	±	13.9	^bc^

^1^ Data are expressed as the mean ± standard deviation (*n* = 6). Different letters (^a^, ^b^, and ^c^) indicate a significant difference at *p* < 0.05 by a one-way ANOVA with Duncan’s post-hoc test.

**Table 4 ijms-22-12547-t004:** Effects of synbiotics supplementation on hepatic cytokines in rats with chronic ethanol feeding ^1,2^.

Groups	TNF-α (pg/μg Protein)				IL-1β (pg/μg Protein)				IL-6 (pg/μg Protein)				IL-10 (pg/μg Protein)			
C	62.0	±	11.8	^ab^	72.0	±	11.0	^ab^	199.6	±	41.3	^ab^	24.1	±	5.4	^ab^
E	67.1	±	18.2	^a^	74.5	±	26.4	^a^	235.6	±	62.8	^a^	27.3	±	7.7	^a^
SC	71.0	±	13.0	^a^	70.0	±	13.7	^ab^	238.8	±	40.7	^a^	23.4	±	4.8	^ab^
PSE	46.7	±	7.5	^b^	44.9	±	9.8	^c^	166.6	±	24.9	^b^	18.9	±	2.0	^b^
ASE	61.2	±	12.0	^ab^	53.2	±	10.6	^bc^	207.4	±	40.3	^ab^	24.7	±	4.8	^ab^

^1^ Data are expressed as the mean ± standard deviation (*n* = 6). Different letters (^a^, ^b^, and ^c^) indicate a significant difference at *p* < 0.05 by a one-way ANOVA with Duncan’s post-hoc test. ^2^ IL, interleukin; TNF, tumor necrosis factor.

**Table 5 ijms-22-12547-t005:** Effects of synbiotics on thiobarbituric acid-reactive substances (TBARSs) in rats with chronic ethanol feeding ^1^.

Groups	Plasma (μM/mL)			Liver (μM/mg Protein)			
C	13.3	±	7.0	0.09	±	0.04	^b^
E	15.5	±	5.5	0.18	±	0.10	^a^
SC	11.3	±	3.9	0.16	±	0.08	^ab^
PSE	13.8	±	6.6	0.11	±	0.01	^b^
ASE	12.8	±	6.2	0.10	±	0.04	^b^

^1^ Data are expressed as the mean ± standard deviation (*n* = 6). Different letters (^a^ and ^b^) indicate a significant difference at *p* < 0.05 by a one-way ANOVA with Duncan’s post-hoc test.

**Table 6 ijms-22-12547-t006:** Effects of synbiotics supplementation on grip strength (g) in rats with chronic ethanol feeding ^1^.

Groups	Initial						Final					
	Grip Strength (g)			Grip Strength/kg BW			Grip Strength (g)			Grip strength/kg BW		
C	932.2	±	128.1	2.8	±	0.4	1135.9	±	349.6	2.7	±	0.7
E	954.4	±	252.9	3.0	±	0.8	1031.8	±	197.1	2.6	±	0.5
SC	996.0	±	128.1	3.1	±	0.4	1108.7	±	263.2	2.7	±	0.7
PSE	1095.1	±	135.9	3.3	±	0.3	994.2	±	177.4	2.5	±	0.5
ASE	1081.9	±	122.1	3.4	±	0.3	1148.0	±	130.4	2.9	±	0.3

^1^ Data are expressed as the mean ± standard deviation (*n* = 6). Different letters indicate a significant difference at *p* < 0.05 by a one-way ANOVA with Duncan’s post-hoc test. BW, body weight.

**Table 7 ijms-22-12547-t007:** The components of synbiotic powder ^1,2^.

Component	Amount/1.5 g
Calories	4 kcal
Protein	1 g
Fat	0 g
Carbohydrate	80 mg
Sodium	14 mg
Inulin from chicory, powdered extract (root)	50 mg
Vitamin B_1_	75 mg
Vitamin B_2_	0.85 μg
Niacinamide	1 mg
Vitamin B_6_	100 μg
Vitamin B_12_	0.3 μg
Biotin	15 μg
Folate	20 μg
Pantothenic acid	0.5 mg
Proprietary blend culture count	2 × 10^9^ CFU
*Lactobacillus acidophilus* and *Lactobacillus bulgaricus*	5.4 × 10^8^ CFU
*Bifidobacterium bifidum* and *Bifidobacterium longum*	1.3 × 10^9^ CFU
*Streptococcus thermophilus*	1.2 × 10^8^ CFU

^1^ The synbiotics FloraGuard^®^ will be provided by Viva Life Science, Costa Mesa, CA, USA. ^2^ CFU, colony forming unit.

**Table 8 ijms-22-12547-t008:** List of antibodies for analyzing related proteins in this study.

Sample	Antibody	Source
Liver	Rabbit polyclonal CYP2E1	Millipore, Burlington, MA, USA
Muscle	Rabbit polyclonal mTOR	Cell Signaling, Danvers, MA, USA
Muscle	Rabbit polyclonal p-mTOR ^Ser2448^	Cell Signaling, Danvers, MA, USA
Muscle	Rabbit polyclonal S6K	Affinity Bioscience, Cincinnati, OH, USA
Muscle	Rabbit polyclonal p-S6K ^T389^	ABclonal Technology, Woburn, MA, USA
Muscle	Rabbit polyclonal Myostatin	Proteintech Group, Inc., Rosemont, IL, USA
Muscle	Rabbit monoclonal MuRF-1	Abcam, Cambridge, UK
Muscle	Rabbit monoclonal Beclin1	Genetex, Irvine, CA, USA
Muscle	Rabbit monoclonal p62	Cell Signaling, Danvers, MA, USA
Muscle	Rabbit monoclonal LC3B II/I	Cell Signaling, Danvers, MA, USA
Internal control	Mouse monoclonal GAPDH	Millipore, Burlington, MA, USA

CYP2E1, cytochrome P450 2E1; TLR4, Toll-like receptor 4; LC3B, microtubule-associated proteins 1A/1B light chain 3B; mTOR, mammalian target of rapamycin; S6K, ribosomal protein S6 kinase; MuRF-1, muscle ring-finger protein-1.

**Table 9 ijms-22-12547-t009:** List of primer sequences for analyzing related mRNA in this study.

	Forward 5′→3′	Reverse 5′→3′
ZO-1	CTTGCCACACTGTGACCCTA	ACAGTTGGCTCCAACAAGGT
Occludin	CTGTCTATGCTCGTCATCG	CATTCCCGATCTAATGACGC
Claudin-1	AAACTCCGCTTTCTGCACCT	TTTGCGAAACGCAGGACATC
β-actin	CACCAGTTCGCCATGGATGACGA	CCATCACACCCTGGTGCCTAGGGC

ZO-1, zonula occludens-1.

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
