# Peer review of "Synbiotics Alleviate Hepatic Damage, Intestinal Injury and Muscular Beclin-1 Elevation in Rats after Chronic Ethanol Administration"

_ijms, 2021, doi:10.3390/ijms222212547_

Round 1

Reviewer 1 Report

Chen, et. al. report on chronic ethanol  studies conducted  in rats fed control or ethanol-containing diets either with or without  inulin-containing synbiotics. The authors measured  parameter of sskeletal muscle protein metabolism and a number of ethanol-induced changes in  livers and intestines of these animals. Qualitative and quantitative analyses of fecal microbiota were also assessed. The authors report that inclusion of inulin-containing synbiotics in the diets ameliorated ethanol-induced injury to these three organs. They attribute normalization of skeletal muscle metabolism in synbiotic-supplemented ethanol-fed rats was due to strengthening of intestinal tight junctions. which allowed limited circulating endotoxin into the circulation. They also postulate that plasma ammonia levels were normalized, as was amino acid metabolism, as judged by the plasma amino acid levels.

Strengths: The paper is generally clear, as are the data, which are presented in both tabular and graphic form.  Data mostly agree with the conclusions.

Weaknesses:  The title is somewhat misleading, as  multiple liver and gut parameters were measured as well as in muscle. Suggest the alternate title: "Synbiotics alleviate hepatic, intestinal and skeletal muscle injury after chronic ethanol administration"

The abbreviations SC, PSE and ASE  are not immediately intuitive. What do they stand for? 

It's not clear what food efficiency means. Please clarify.

The Beclin-1 levels  appear to be the only parameter used to measure protein catabolism . Authors should also measure proteasome chymotrypsin-like  activity and/or the levels of LAMP -1 (lysosome content). The levels of P62 and LC3II suggest that ethanol feeding slows muscle protein catabolism. 

Were all diets isocaloric? Synbiotics contain 4 Cal/1.5 g. (Table 7).

Some entries in the supplementary table with the amino acid levels seem inconsistent with the other entries. Authors need to re-check their accuracy.

Reviewer 2 Report

The paper “: Effects of inulin-containing synbiotics on muscle protein synthesis and degradation in rats chronically fed with alcohol" investigates the effect of inulin-containing synbiotics on to counteract liver damage and muscle loss in rats with chronic ethanol feeding, in a likely relationship with the ability of symbiotics to maintain intestinal health.  English is good and data convincing and properly presented and interpreted. Only a remark is given, as detailed below, which needs attention to make the manuscript to need a minor revision to become suitable for publication in IJMS. 

Abstract: lines 20 -21, description of the groups of animals is to be revised, according to following text and related Figure legends. A table reporting groups and time/treatments, maybe at the beginning of results, should be helpful to make them clearer to the reader.  This could also allow to remove repeated information where the case in legends to Tables and Figures.

Round 2

Reviewer 1 Report

 The manuscript has been improved and the authors have made improvements. I  recommend that the paper be accepted,  but the authors need to  "downplay" (i.e. de-emphasize) their muscle data, as the liver and gut results in these animals more strongly indicate a beneficial effect of synbiotics, which makes sense. The effect on skeletal muscle, which is more "distal", was not so obvious. It would also be helpful, (but not mandatory) that the authors report the serum ethanol  levels in these animals. Such data would be valuable, as the inclusion of synbiotics may have enhanced first pass ethanol metabolism, resulting in the alleviation of tissue injury.   

The title of the paper has been revised and  more accurately  reflects its content. 
